# Spatial distribution of animal source food consumption and associated factors among children aged 6–23 months in Ethiopia: A geographically weighted regression analysis

**Mekuriaw Nibret Aweke**[1]*, **Amare Mesfin**[1], **Gebrie Getu Alemu**[2], **Berihanu Mengistu**[1], **Tewodros Getaneh Alemu**[3], **Habtamu Wagnew Abuhay**[2]

1 Department of Human Nutrition, Institute of Public Health, College of Medicine and Health Sciences, University of Gondar, Gondar, Ethiopia, 2 Department of Epidemiology and Biostatistics, institute of public health, College of Medicine and health science, University of Gondar, Gondar, Ethiopia, 3 Department of Pediatrics and Child Health Nursing, School of Nursing, College of Medicine and Health Sciences, University of Gondar, Gondar, Ethiopia

* mekunib@gmail.com

## Abstract

### Introduction

Optimal nutrition during early childhood is essential for growth, cognitive development, and overall health. Animal source foods(ASF) provide essential nutrients like high-quality protein, iron, zinc, calcium, and vitamin B12, which are vital for the physical and cognitive development of young children and to reduce the burden of malnutrition. In Ethiopia malnutrition among children under five remains a significant public health issue. Many children are suffering from chronic and acute undernutrition. This study provides the first spatial analysis of animal source foods consumption among children aged 6–23 months.

### Objectives

The aim of this study is to investigate the spatial distribution of ASF consumption and its associated factors among children aged 6–23 months in Ethiopia using data from the Ethiopia miniDHS 2019.

### Methodology

A cross-sectional study design was used using mini-DHS 2019 data among children aged 6–23 months. Descriptive statistics were used to summarized the study population characteristics and ASF consumption prevalence. Spatial analysis techniques, including Geographical mapping and Moran's I statistic assessed the spatial distribution of ASF consumption. Geographically weighted regression analyses identified socio-economic, demographic, and geographic factors associated with ASF consumption.

**Data availability statement:** The dataset utilized for this analysis can be accessed on the DHS program website (http://dhsprogram.com). All data produced and analyzed in this study, including maps, tables, and textual information, are included in this article

**Funding:** The author(s) received no specific funding for this work.;

**Competing interests:** The authors have declared that no competing interests exist.

**Abbreviations:** ASF, Animal Source Foods; DHS, Demographic and Health Survey; miniDHS, Mini Demographic and Health Survey; GWR, Geographically Weighted Regression; WHO, World Health Organization; UNICEF, United Nations International Children's Emergency Fund.

## Results

The study found that 47.7% of Ethiopian children aged 6–23 months consume ASF. Children in regions like Amhara, Tigray, Benishangul-Gumuz, western SNNPR, and Gambela are less likely to consume these foods. Factors linked to higher ASF consumption include mothers with more education, smaller families, households following the Orthodox religion, and wealthier families..

## Conclusion and Recommendations

According to this study finding animal source food consumption among Ethiopian children aged 6–23 months is relatively low. The finding revealed the significant regional disparaties of ASF consumptions. Factors associated with ASF consumption include maternal education, household size, wealth, and religion. Efforts should focus on working to increase maternal education, providing and expansion of family planning services, and increasing affordability of ASF through economic improvement of households. In addition strengthening food supply chains and integrating ASF promotion into healthcare are also essential for improving child nutrition. Interventions in low-consumption areas should address specific local needs to effectively boost ASF consumption and improve child nutrition outcomes.

## Introduction

Malnutrition is a major global health issue, especially affecting children aged 6–23 months. Globally, 149 million children under five years suffer from stunted growth, a direct result of inadequate nutrition during their early years [1]. This issue is critical, as malnutrition accounts for an estimated 54% of child mortality worldwide [2]. About 47 million children under five are wasted [3] and 144 million children suffer from stunting and among these 40% are from sub-Saharan Africa [4]. In low- and middle-income countries the prevalence of underweight is 16% in urban settings and 30% in rural areas [5]. The Ethiopian 2019 mini-DHS found that while malnutrition rates have slowly declined over the past decade, under-five children still face high levels of malnutrition. Specifically, 37% of these children are stunted, 7% are wasted, and 21% are underweight [6]. In early childhood, malnutrition reduces educational attainment and work productivity, while also increasing the risk of chronic diseases later in life. These effects limit individual potential and have significant implications for broader socioeconomic development [7]. The causes of malnutrition differ based on geographical context and can change over time [8]. Child malnutrition is primarily caused by inadequate or inappropriate nutritional intake that does not meet the physiological requirements necessary for healthy growth and development [9,10].

Proper nutrition during the first two years of life is crucial for ensuring children's optimal growth, development, and long-term health [11]. The period from 6 to 23 months is a critical window for growth and development, during which children transition from exclusive breastfeeding to complementary feeding [12]. A child's

diet quality should include a diverse range of food components to ensure a comprehensive and balanced nutrition [13]. Among the various components of a child's diet, animal source foods (ASF) such as meat, milk, and eggs play an indispensable role [14,15]. Animal source foods are superior to plant-based foods in terms of micronutrient and protein quality as they provide higher concentrations of essential nutrients with greater bioavailability [16,17]. Among the essential nutrients supplied by animal source foods are iron (Fe), zinc (Zn), calcium (Ca), riboflavin, vitamin A, and vitamin B12. As a result, incorporating even small quantities of ASFs can greatly enhance the adequacy of the overall diet [16,18].

In low- and middle-income countries such as Ethiopia, many children often experience restricted access to and availability of ASFs. Inadequate intake of ASF during early childhood is linked to stunting, anemia and delayed development particularly in low-income settings [19]. Previous study revealed that only 14% of children aged 6–23 months meet minimum dietary diversity requirements, with only 8% consuming meat, fish, or poultry, and about 17% and 25% eating eggs and dairy products, respectively [20]. Another study showed that only 22.7% of children in Ethiopia consume ASFs, indicating that only one in five children has access to ASF consumption [21]. Despite improvements in ASF production in the country, rapidly rising food prices and poor economic access have limited ASF consumption. As a result, many rural households rely predominantly on maize and legumes for their diet [22].Various factors influence ASF consumption. These include women's and child characteristics, religious practices such as fasting, and socio-economic factors such as wealth status [22]. Additionally, a low level of nutrition, knowledge, low health service utilization, availability of livestock's, low educational level of mothers, number of children in the household has also been reported as a factor affecting ASF consumption [21].

To adress undernutrition in children aged 6–23 months, Ethiopia has implemented targeted interventions focusing on promoting optimal feeding practices and addressing micronutrient deficiencies [23]. For instance, the Productive Safety Net Program(PSNPs) enhances household food security, thereby improving young children's nutritional outcomes. Additionally, micronutrient supplementation, including vitamin A and iron, effectively prevents deficiencies. Moreover, community-based nutrition education enhances caregiver knowledge and practices, while improved healthcare services provide vital support and monitoring. Furthermore, the 'Seqota' Declaration, introduced in 2015, aims to eliminate stunting in children under 2 years by 2030 [15]. Despite these comprehensive measures, malnutrition remains a significant public health issue, with regional disparities in dietary practices and access to nutritious foods.

This study seeks to fill a gap in the existing literature, as limited research has been conducted on how spatial and socio-economic factors influence ASF intake in Ethiopia. Examining the spatial distribution of ASF consumption is critical for formulating effective nutrition policies in Ethiopia. This study utilizes Geographically Weighted Regression (GWR) to assess the impact of spatial and socio-economic factors on ASF intake among children aged 6–23 months. Given Ethiopia's diverse socio-cultural and ecological landscape, GWR allows for location-specific analysis, offering more precise insights for targeted interventions. The results will inform the development of targeted interventions aimed at mitigating malnutrition and enhancing child health through context-specific dietary strategies.

## Methodology

### Study design and period

This study utilizes a cross-sectional design based on data from the 2019 Ethiopian Mini Demographic and Health Survey (mini-EDHS) to investigate the spatial patterns of ASF consumption and its associated factors among children aged 6–23 months. The mini-EDHS is a nationally representative survey that provides comprehensive data on various health and demographic indicators, including child nutrition. Following full-scale DHS surveys in 2000, 2005, 2011, and 2016, the 2019 mini-EDHS provides essential trend data and aids policymakers in shaping health programs. The survey was conducted from March 21, 2019, to June 28, 2019, using a nationally representative sample that allowed for estimates at national, regional, urban, and rural levels [24].

## Study setting

The study was conducted in Ethiopia, a strategically located country in the Horn of Africa, situated between latitudes 3° and 14.8° N and longitudes 33° and 48° E. It shares borders with Somalia, Sudan, Djibouti, Kenya, and Eritrea, extending over a total border length of 5,311 km. Ethiopia is the 10th largest country in Africa and the second most populous, with a population exceeding 115 million [25]. Ethiopia is administratively divided into nine regions: Tigray, Afar, Amhara, Oromia, Somalia, Benishangul-Gumuz, Southern Nations, Nationalities, and Peoples Region (SNNPR), Gambela, and Harari, as well as two self-administered cities, Addis Ababa and Dire Dawa.

## Source and study population

The 2019 mini-EDHS employed a two-stage stratified sampling approach. In the first stage, 305 enumeration areas (EAs) were selected, with 93 in urban and 212 in rural areas, using probability proportional to size. To manage large EAs, which had more than 300 households, they were segmented, and only one segment was chosen for the survey. In the second stage, 30 households were systematically selected from each segment [24]. This design ensured comprehensive coverage and precise data collection across different regions. Since the probability of selecting each household was not uniform, sample weighting was used in the analysis. Total weighted samples of 1,460 children aged 6–23 months who were the youngest and lived with their mother were included in the analysis (Fig 1).

## Variables

### Dependent variable

**Animal Source Foods Consumption:** Consumption of animal sources was considered if a child aged 6–23 months, living with their mother, reported eating at least one of the following foods within the 24 hours prior to the interview: 1) eggs, 2) meat (beef, lamb, goat, or chicken), 3) fresh or dried fish or shellfish, 4) organ meats such as heart or liver, and 5) milk and milk products. Children were divided into two groups according to their consumption of ASF. Children who consumed ASF were given a value of 1, while those who did not were given a value of 0. Finally, a weighted proportion of ASF per cluster was utilized for spatial analysis.

### Independent variables

**Sociodemographic characteristics.** The independent variables examined in this study were selected from a review of existing literature. These variables include child related factors, maternal, and community characteristics.. From child characteristics age was divided into four groups: 6–8 months, 9–11 months, 12–17 months, and 18–23 months, allowing for a detailed analysis of different developmental stages and nutritional needs.

The respondent's characteristics include age, educational status, marital status, occupation, total number of children in the household, religion and place of residence. Age is categorized into three groups: 15–24 years, 25–34 years, and 35–49 years, to account for generational differences in knowledge and practices related to child nutrition. Educational status was categorized as no education, primary, secondary, and higher education or above.

Marital status is divided into married and not married to examine the impact of family structure on child nutrition..

The type of place of residence, whether urban or rural, is also included to capture the environmental context in which the child is raised. This factor is important as it affects access to resources, healthcare services, and nutritional options, which in turn influence ASF consumption.

**Socioeconomic factors.** The wealth index in this study classifies households into five categories based on their economic status: Poorest, Poor, Middle, Rich, and Richest. This classification is derived from a range of assets and living conditions, including ownership of livestock, household items, and access to basic services such as electricity and sanitation. Thus, the wealth index can significantly influence the quality and variety of the diet, including the consumption of ASF, by reflecting the household's ability to access and afford these important nutritional resources.

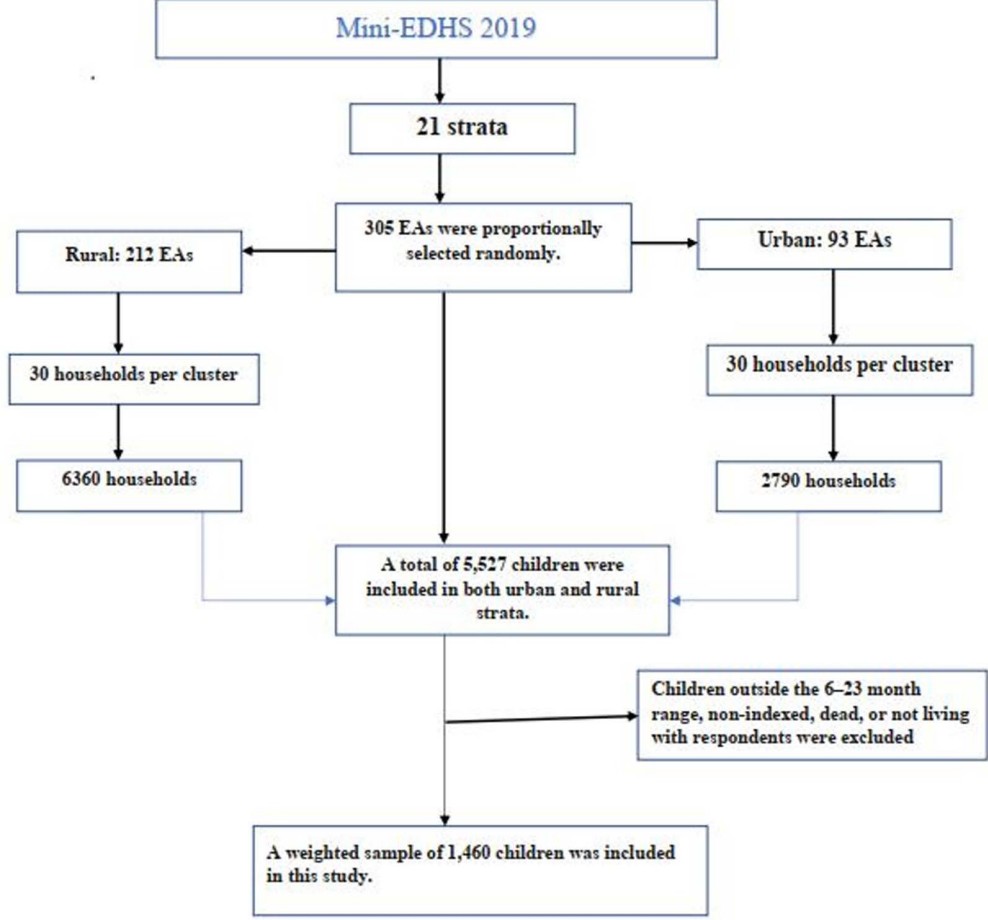

**Fig 1. Schematic representation of the sampling procedure used in the study of animal source food (ASF) consumption among children aged 6–23 months in Ethiopia, 2019.**

   **Health service-related factors.** Factors related to birth and delivery include the birth order of the index child, whether the child was delivered via cesarean section, antenatal follow-up, postnatal follow-up, and whether the respondent received counseling about breastfeeding from health workers. These factors are crucial in understanding the healthcare interactions that may affect child feeding practices and the consumption of ASF.

## Data collection and tools

The data for the analysis of ASF consumption were collected through face-to-face interviews using structured questionnaires at both individual and household levels. During the data collection period, mothers of children aged 6–23 months were asked to provide detailed information on various factors related to ASF consumption. This included socio-demographic details, socioeconomic status, health service utilization, and child-specific characteristics. The goal was to gather comprehensive data on how these factors influence the consumption of ASF among young children in Ethiopia.

## Data management and analysis

To analyze ASF consumption, we first tabulated the proportions of both ASF and potential predictor variables using STATA, then exported the data to Excel for further examination. First we adjusted the data for ASF consumption and

explanatory variables using sample weights. For understanding spatial patterns, we used ArcGIS 10.7 and SaTS-can V.9.6 software to detect local clusters. Records with missing values in the outcome variable (ASF consumption) were excluded, as per DHS recommendations to retain only cases with complete outcome information. For independent variables, a complete case analysis was used since the proportion of missing data was small. Sampling weights (v005/1,000,000) were applied to adjust for the multi-stage sampling design and ensure representativeness, following DHS guidelines.

### Global spatial autocorrelation

To assess the spatial distribution of ASF consumption, we used ArcGIS 10.7(ESRI Inc., Redlands, CA, USA, version 10.7) software. For spatial autocorrelation and to identify hot spot areas. We applied the Global Moran's I statistic to determine whether ASF consumption among children aged 6–23 months was dispersed, clustered, or randomly distributed across Ethiopia. Moran's I values close to -1 indicate a dispersed pattern, values close to +1 suggest a clustered pattern, and a value around 0 reflects a random distribution. The identification of hot spots and cold spots for ASF consumption was further refined using z-scores and significant p-values from the hot spot analysis (Getis-Ord Gi*).

### Spatial interpolation

To estimate ASF consumption in unsampled areas, we used spatial interpolation techniques. The Ordinary Kriging Gaussian interpolation method was employed to predict ASF intake among children. This method helps minimize prediction uncertainty and filter out measurement errors. We constructed a semi-variogram model based on input data from sampled locations to determine weights for predicting new values in unsampled areas, resulting in a simulated semi-variogram model.

### Spatial scan statistics

For local cluster detection, we used SaTScan version 10.1 software [26]. We applied a purely spatial Bernoulli-based model to identify statistically significant clusters with high or low rates of ASF consumption. Children who consumed ASF within 24 hours were considered cases, while those who did not were controls. The Bernoulli model scanned the study area with a moving window, excluding regions outside the study bounds. We set the maximum spatial cluster size to less than 50% of the population to detect both small and large clusters, discarding those exceeding this limit. Each potential cluster was assessed using a log-likelihood ratio test to determine if the number of observed cases was significantly higher than expected. The cluster with the highest likelihood ratio was identified as the primary cluster. Primary and secondary clusters were then ranked and assigned p-values based on the likelihood ratio test results from 999 Monte Carlo simulations.

### Spatial regression

**Ordinary Least Squares (OLS).** To understand the factors influencing ASF consumption across the study area, we used spatial regression modeling. The Ordinary Least Squares (OLS) model provided a broad overview by estimating a single coefficient for each explanatory variable throughout the entire region. For the OLS results to be reliable, several key conditions had to be met. These included having statistically significant coefficients with the correct direction, ensuring the model was non-stationary, including all important variables, and avoiding multicollinearity. Additionally, the residuals needed to be normally distributed and free from any spatial patterns or autocorrelation.

We used a data mining tool to select a model that met these OLS requirements and demonstrated high adjusted R² values. The final model was validated through internal cross-validation, and we checked for multicollinearity (with a variance inflation factor of less than 7.5) to ensure that the independent variables were not redundant.

## Geographically weighted regression analysis

A predictor that is influential in one area might not be as significant in another. This variation, known as non-stationarity, can be detected using GWR. Unlike OLS, which applies one regression equation to the entire study area, GWR creates a separate equation for each cluster. As a result, the coefficients in GWR vary from one cluster to another, providing a more nuanced understanding of how different factors affect ASF consumption in different regions. The GWR map of the coefficients of each predictor variable guides targeted interventions.

$$yi = B_0(uivi) + \sum_{k=1}^{p} Bk\ (ui,\ vi)xik + \varepsilon i$$

where yi is the observation of response; (uivi) is the latitude and longitude; βk (ui, vi) (k = 0, 1,… p) is the p unknown function of the geographical location (uivi); xik is the independent variable at location (uivi), where i is equal to 1,. 2,…; and εi is the error term/residual with zero mean and homogeneous variance σ.

## Ethical consideration

Permission for data access was obtained from a major Demographic and Health Survey through an online request at (http://www.dhsprogram.com). The data used for this study were publicly available with no personal identifier. Our study was based on secondary data from Ethiopian Demographic and Health Survey and we have secured the permission letter from the main Demographic Health and Survey.

# Results

## Characteristics of the respondents and study children

The study included weighted sample of 1,460 children aged 6–23 months. More than half of these children were boys 763(52.27%). The majority 1,047(71.76%) lived in rural areas while about a quarter resided in urban areas 412(28.24%). Most mothers 724(49.61%) were between 25 and 34 years old and nearly a third 471(32.23%) were aged 15–24 years. Regarding education, nearly half of the mothers 649(44.47%) had no formal education and about 609(41.68%) had only primary schooling. A significant majority of mothers were married 1,377(94.37%). In terms of wealth, approximately one-fifth of the children324(22.22%) came from the richest households while nearly a quarter 309(21.18%) were from poor households. The majority of mothers 1,114(76.33%) received antenatal care during their pregnancy with their youngest child.. Regionally, Oromia had the highest representation with 542 children (37.10%), followed by Amhara with 326 (22.35%) and SNNPR with 296 (20.28%) Table 1.

## Prevalence of ASF consumption

In this study, the overall prevalence of ASF consumption was 47.7% (95% CI: 45.1% - 50.2%). Regional variations were observed across Ethiopia. The highest prevalence was reported in Oromia at 20.88%, followed by the Amhara region at 6.16%. Animal source food consumption was more common among mothers of child aged 6–23 months living in rural areas, with a prevalence of 33.45%, compared to 14.20% in urban areas. Mothers with 1–2 children had a higher prevalence of ASF consumption compared to those with 3–5 children. In addition ASF consumption among children was higher for those whose mothers had antenatal care (ANC) during the last pregnancy.

## Spatial autocorrelation of ASF consumption among children aged 6–23 months in Ethiopia

To assess the spatial clustering of ASF consumption, we conducted a global spatial statistics analysis using Moran's I value. The results showed that ASF consumption among children aged 6–23 months in Ethiopia is

**Table 1. Sociodemographic and Health Service-Related Characteristics of Mothers and Children Aged 6–23 Months in Ethiopia, 2019 (n = 1,460).**

| Variable | Categories | Frequency (n) | Percentage (%) |
|---|---|---|---|
| **Sex of the child** | Male | 763 | 52.2 |
| | Female | 697 | 47.8 |
| **Age of the mother** | 15–24 | 471 | 32.2 |
| | 25–34 | 724 | 49.6 |
| | 35–49 | 265 | 18.2 |
| **Place of residence** | Urban | 412 | 28.3 |
| | Rural | 1048 | 71.7 |
| **Educational status** | No education | 649 | 44.5 |
| | Primary education | 609 | 41.7 |
| | Secondary education | 120 | 8.2 |
| | Higher education and above | 82 | 5.6 |
| **Marital status** | Married | 1377 | 94.3 |
| | Not married | 82 | 5.7 |
| **Wealth index level** | Poorest | 294 | 20.1 |
| | Poor | 309 | 21.2 |
| | Middle | 275 | 18.8 |
| | Rich | 257 | 17.6 |
| | Richest | 324 | 22.3 |
| **Age of infant** | 6–8 months | 255 | 17.5 |
| | 9–11 months | 221 | 15.1 |
| | 12–17 months | 551 | 37.8 |
| | 18–23 months | 433 | 29.6 |
| **Religion** | Orthodox | 537 | 36.8 |
| | Catholic | 9 | 0.6 |
| | Protestant | 412 | 28.3 |
| | Muslim | 474 | 32.5 |
| | Other | 29 | 2.0 |
| **Number of children** | 1–2 Children | 1307 | 89.5 |
| | 3–5 Children | 154 | 10.5 |
| **ANC follow-up** | No ANC | 345 | 23.6 |
| | Had ANC | 1114 | 76.4 |
| **Place of delivery** | Home | 655 | 44.8 |
| | Health Facility | 804 | 55.2 |
| **Delivery by Caesarean** | No | 1370 | 93.8 |
| | Yes | 90 | 6.2 |
| **Counseling by Health** | No | 336 | 23.0 |
| | Yes | 779 | 53.3 |
| **Birth order** | First to Third Child | 880 | 60.3 |
| | Fourth Child or More | 579 | 39.7 |
| **Region** | Tigray | 103 | 7.1 |
| | Afar | 20 | 1.4 |
| | Amhara | 326 | 22.3 |
| | Oromia | 542 | 37.1 |
| | Somali | 89 | 6.1 |
| | Benishangul | 17 | 1.2 |
| | SNNPR | 296 | 20.3 |
| | Gambela | 7 | 0.5 |
| | Harari | 4 | 0.3 |
| | Addis Ababa | 48 | 3.3 |
| | Dire Dawa | 9 | 0.6 |

significantly clustered across different regions. Specifically, we found a Global Moran's I of 0.39, a Z-score of 6.85, and a p-value less than 0.001. This suggests that the patterns of ASF consumption are not randomly distributed but are instead spatially interdependent, with certain areas exhibiting higher rates of ASF consumption. The figures below illustrate these clustered patterns, providing a visual representation of regions where ASF consumption is more prevalent (Fig 2).

The incremental spatial autocorrelation evidenced that with varying distance bands starting from 200,000 meters, significant clustering of ASF consumption among children aged 6–23 months in Ethiopia was detected at distances of approximately 300,000 meters and 400,000 meters. Statistically, the most pronounced spatial clustering was observed at 400,000 meters, as indicated by the highest Z-score (Fig 3)

### Hot spot analysis

We used the Getis-Ord Gi* Spatial Statistics method to conduct a Hot Spot Analysis and uncover significant patterns in ASF consumption across Ethiopia. As shown in Fig 4 below, there are clear patterns of both low (cold spot) and high (hot spot) ASF consumption areas among children aged 6–23 months in Ethiopia. This analysis revealed clear geographic

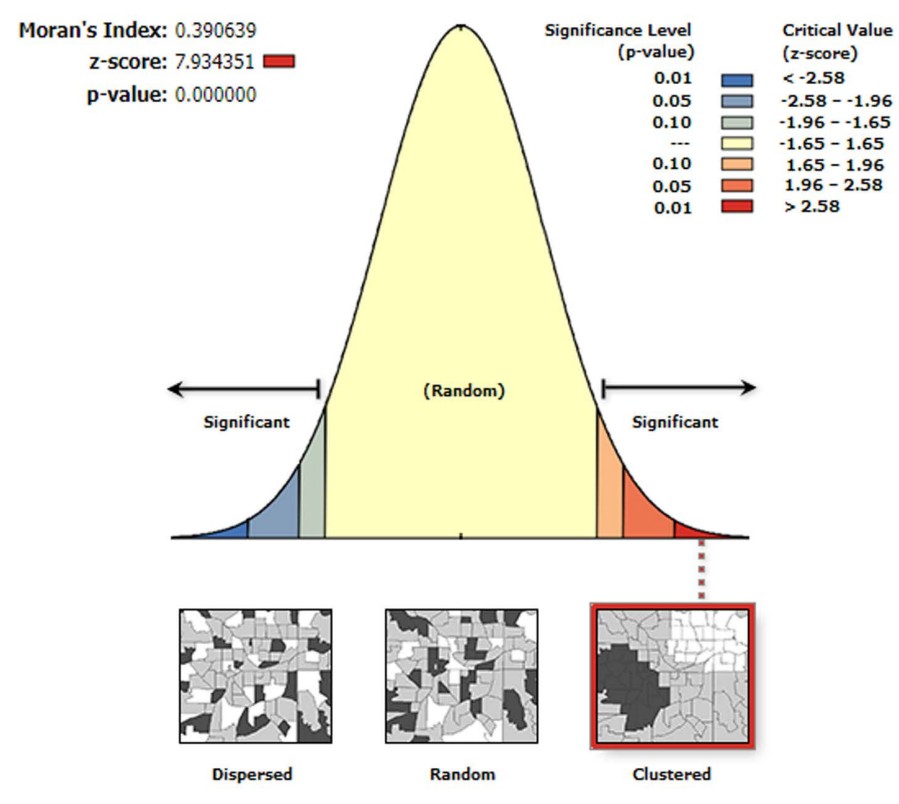

Given the z-score of 7.93435098562, there is a less than 1% likelihood that this clustered pattern could be the result of random chance.

**Fig 2. Spatial autocorrelation report of animal source food (ASF) consumption among children aged 6–23 months in Ethiopia, 2019.**

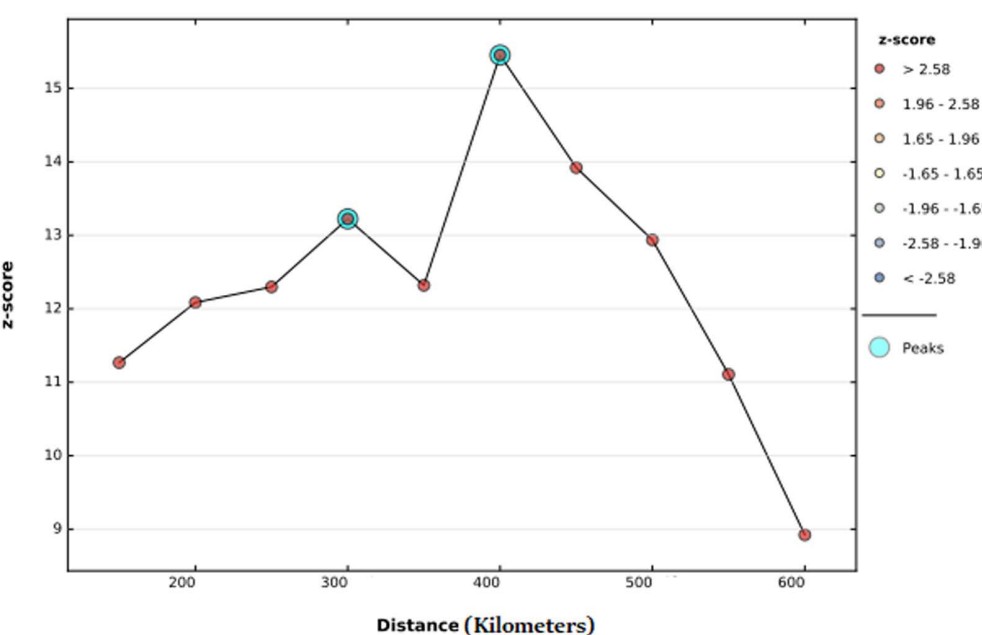

**Fig 3. Spatial autocorrelation by distance for animal source food (ASF) consumption among children aged 6–23 months in Ethiopia, 2019.**

clusters where ASF consumption is notably high or low. For example, areas like Amhara, Tigray, Benishangul-Gumuz, western SNNPR, and Gambela region show significant clustering of low ASF consumption among children aged 6–23 months (Fig 4).

### Animal source food consumption cluster and outlier (Anselin local Moran's I) analysis

We conducted an analysis of ASF consumption in Ethiopia using Anselin Local Moran's I spatial statistics to identify significant clusters and outliers. This method helps us pinpoint areas where high or low ASF consumption is either concentrated or stands out from surrounding regions. The map categorizes regions into several groups. High-High Clusters represent areas where ASF consumption is notably high, and these high rates are surrounded by other regions with similarly high consumption. These clusters are prominent in eastern and southern Somali, central Oromia, Addis Ababa, Benishangul-Gumuz, southern Amhara, and central Afar.

In contrast, Low-Low Clusters identify areas with low ASF consumption that are surrounded by other regions with similarly low rates. Significant low-low clusters are found in western SNNPR, northern and eastern Somali, central Afar, Northern Tigray, Amhara, Addis Ababa and Benishangul-Gumuz. This detailed spatial analysis provides valuable insights into ASF consumption patterns across Ethiopia and helps identify regions that may benefit from targeted interventions and policy efforts (Fig 5).

### Spatial interp\olation using kriging

In this study, we employed an Empirical Bayesian Kriging interpolation method to estimate ASF consumption. The results indicated that children aged 6–23 months across most parts of Ethiopia were vulnerable to low ASF consumption.

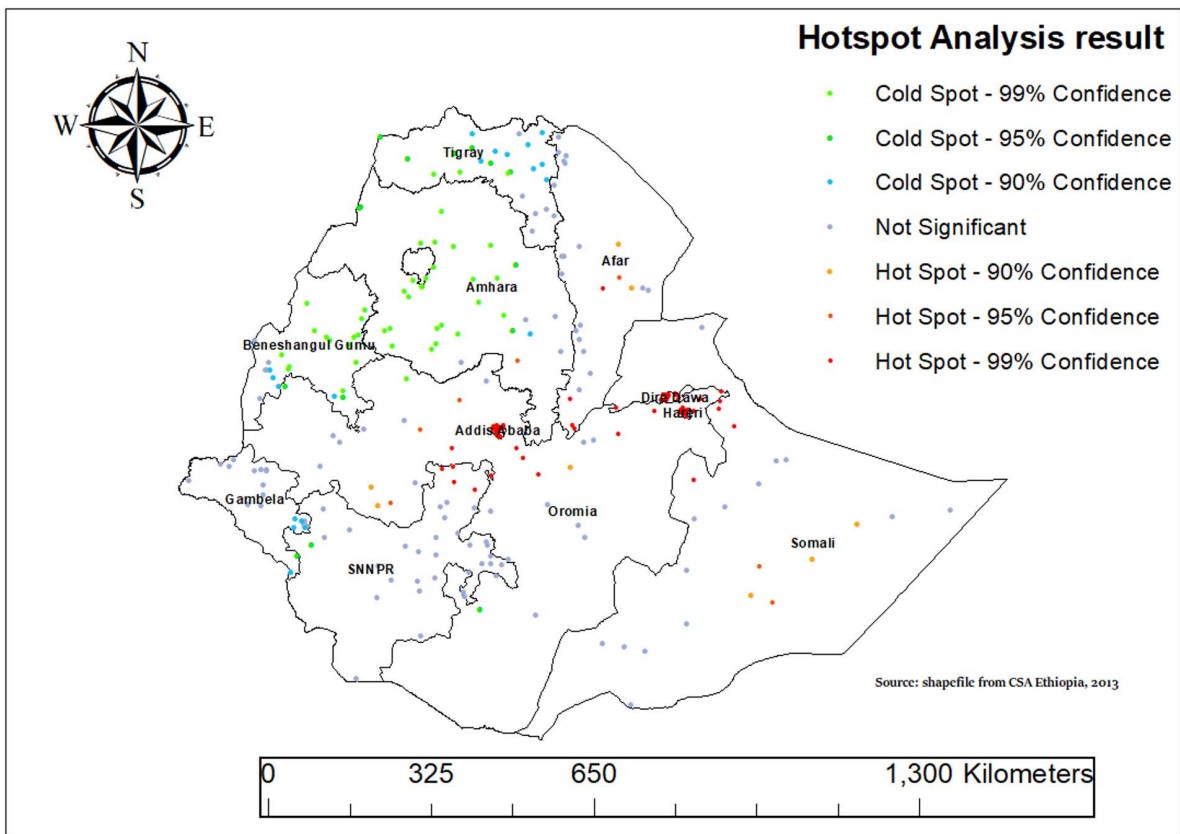

**Fig 4. Hotspot analysis of animal source food (ASF) consumption among children aged 6–23 months in Ethiopia, 2019.**

However, relatively higher levels of ASF consumption were observed in the Somali, Dire Dawa, Hareri, Afar, Addis Ababa, northern SNNPR and Oromia regions (Fig 6).

## Spatial SaTScan analysis

The SaTScan cluster analysis identified significant spatial clusters of ASF consumption among children aged 6–23 months in Ethiopia. The SaTScan method, using the Bernoulli model for spatial analysis, identified clusters with significantly high or low rates of ASF consumption. The analysis revealed a significant spatial cluster of ASF consumption patterns among children aged 6–23 months across Ethiopia. The primary clusters were located at coordinates 7.595318 N, 42.898767 E, covering an area with a 468.49 km radius in Eastern Ethiopia including Somali, Oromia, Hareri, Dire Dawa and SNNR. Children aged 6–23 months living in the primary clusters were 44% more likely to consume animal source ASF compared to those outside these areas (RR = 1.44, 19.641978, P-value < 0.001) (Table 2 and Fig 7).

## Spatial regression results

**OLS analysis result.** In the OLS analysis, the joint F- and Wald-statistics parameters were significant with p-value <0.001 indicating that the model was statistically significant. This model explains 21% of ASF consumption spatial variability. Multicollinearity wasn't an issue during analysis as indicated by VIF reports < 7.5 (Table 3). The Jarque-Bera test for normality of residuals was not significant (p = 0.09), suggesting that the residuals are normally distributed. However, the Koenker test indicated a statistically significant result (p < 0.001), revealing a non-stationary relationship between

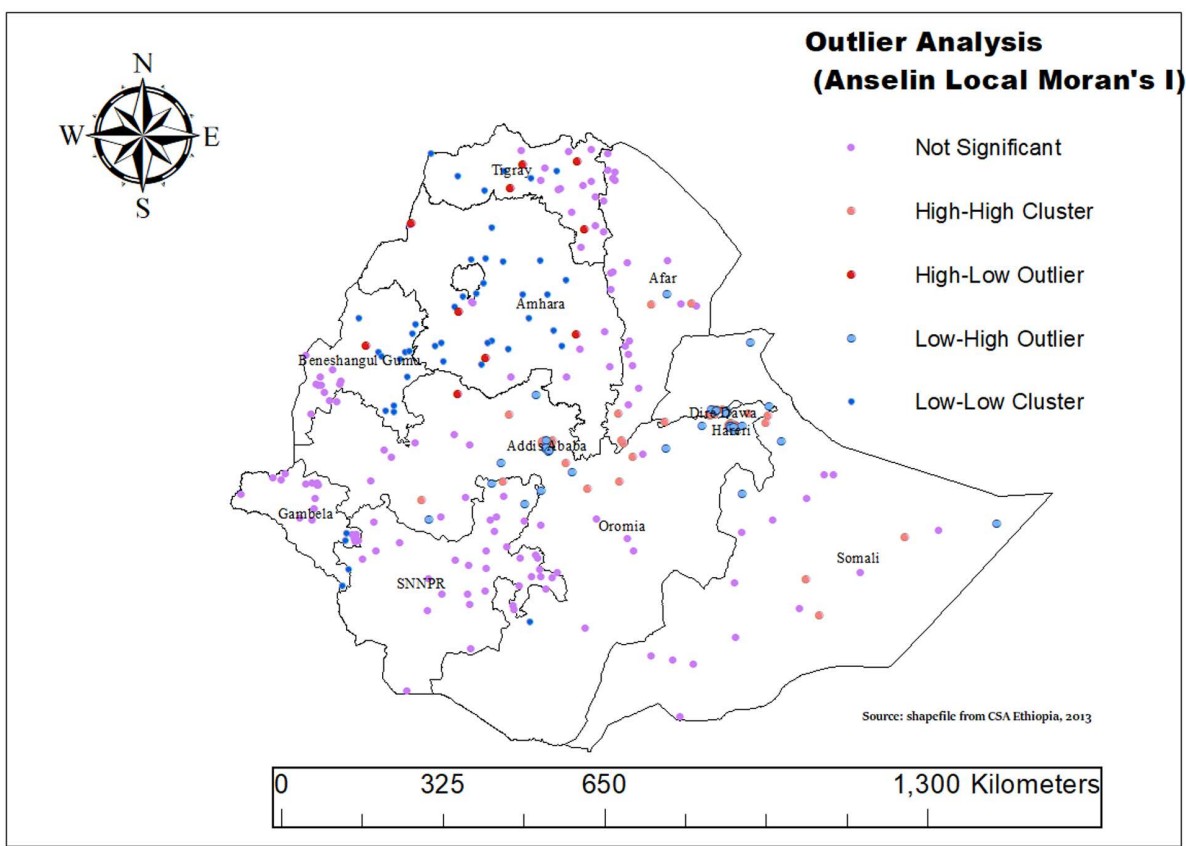

**Fig 5. Outlier and hotspot analysis of animal source food (ASF) consumption among children aged 6–23 months in Ethiopia, 2019.**

the predictor variables and ASF consumption. This implies that a local coefficient for each explanatory variable should be considered to better understand the relationship. Households with two or fewer children, those in the highest wealth index, mothers with secondary or higher education, and a higher proportion of Orthodox religion followers are significantly associated with higher ASF consumption.

## Geographically Weighted Regression (GWR) analysis

The Ordinary Least Squares (OLS) regression analysis initially predicted hotspot areas for ASF consumption. However, as a global model, OLS assumes a stationary relationship between each predictor variable and ASF across the entire study area, which may not capture variations specific to different clusters. This limitation is highlighted by the significant Koenker statistics (p < 0.001) observed in the study, indicating a violation of the OLS assumption of stationary independent variables. In contrast, the GWR model, a local model, effectively addresses these cluster-specific variations.

The GWR analysis demonstrated notable improvements over the OLS model. The adjusted R² increased from 26 in the OLS model to 32 in the GWR model, indicating that GWR better explained the geographical variation in ASF consumption among children aged 6–23 months in Ethiopia. Additionally, the GWR model achieved a lower AICc value of 110.67 compared to the OLS model's AICc of 140.89, reflecting an enhanced fit for examining ASF consumption patterns (Table 4).

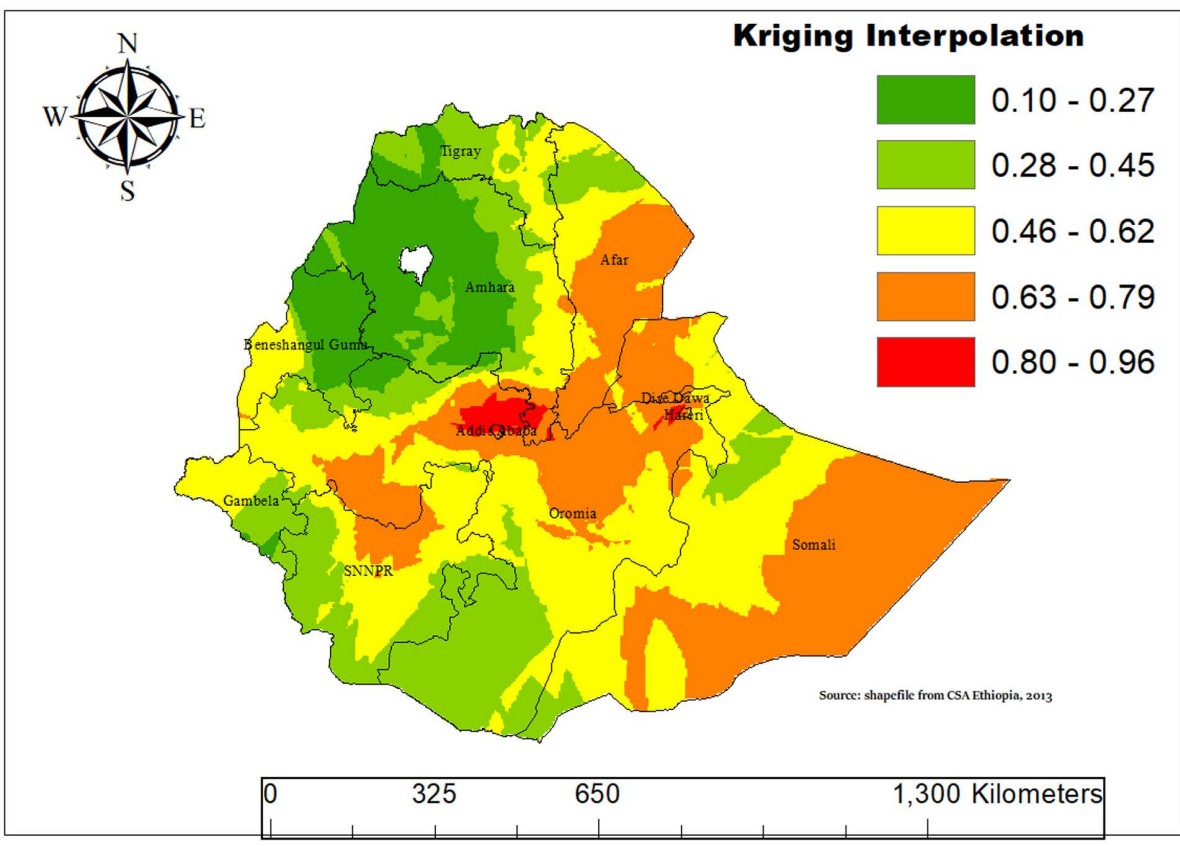

**Fig 6. Spatial interpolation map showing the distribution of animal source food (ASF) consumption among children aged 6–23 months in Ethiopia, based on 2019 data.**

**Table 2. Significant clusters of ASF consumption among children aged 6-23 months old in Ethiopia, 2019.**

| Cluster Type | Location IDs Included | Coordinates/ Radius(km) | Population | Cases | RR | LLR | P-value |
|---|---|---|---|---|---|---|---|
| Primary | 145, 134, 131, 133, 123, 122, 136, 132, 129, 138, 121, 250, 107, 248, 135, 249, 255, 247, 244, 254, 245, 252, 241, 234, 233, 243, 242, 235, 239, 237, 246, 240, 236, 231, 238, 232, 130, 253, 137, 251, 109, 128, 300, 108, 299, 303, 301, 302, 298, 304, 142, 295, 294, 293, 296, 292, 305, 286, 287, 288, 290, 291, 289, 285, 297, 284, 283, 282, 281, 106, 140, 111, 88, 127, 110, 105, 102, 141, 124, 28, 41, 125, 103, 126, 143, 40, 104, 42, 43, 69, 90, 50, 101, 114, 144, 49, 117, 139, 68, 183, 32, 48, 47, 116, 26 | 7.595318 N, 42.898767 E/ 468.49 | 360 | 223 | 1.44 | 19.64 | 0.001 |
| Secondary | 96, 91, 97, 195, 95, 204, 194, 196, 191, 201, 179, 189, 180, 177, 94, 174, 223, 120, 222, 221, 210, 190 | 7.621217 N, 36.610611 E/ 151.45 | 130 | 92 | 1.56 | 15.583436 | 0.001 |
| Tertiary | 260, 261, 259, 276, 263, 275, 258, 265, 262, 257, 264, 270, 267 | 9.001887 N, 38.709463 E/ 8.48 | 28 | 25 | 1.91 | 11.117113 | 0.004 |

**RR: Relative Risk; LLR: Log-Likelihood Ratio**

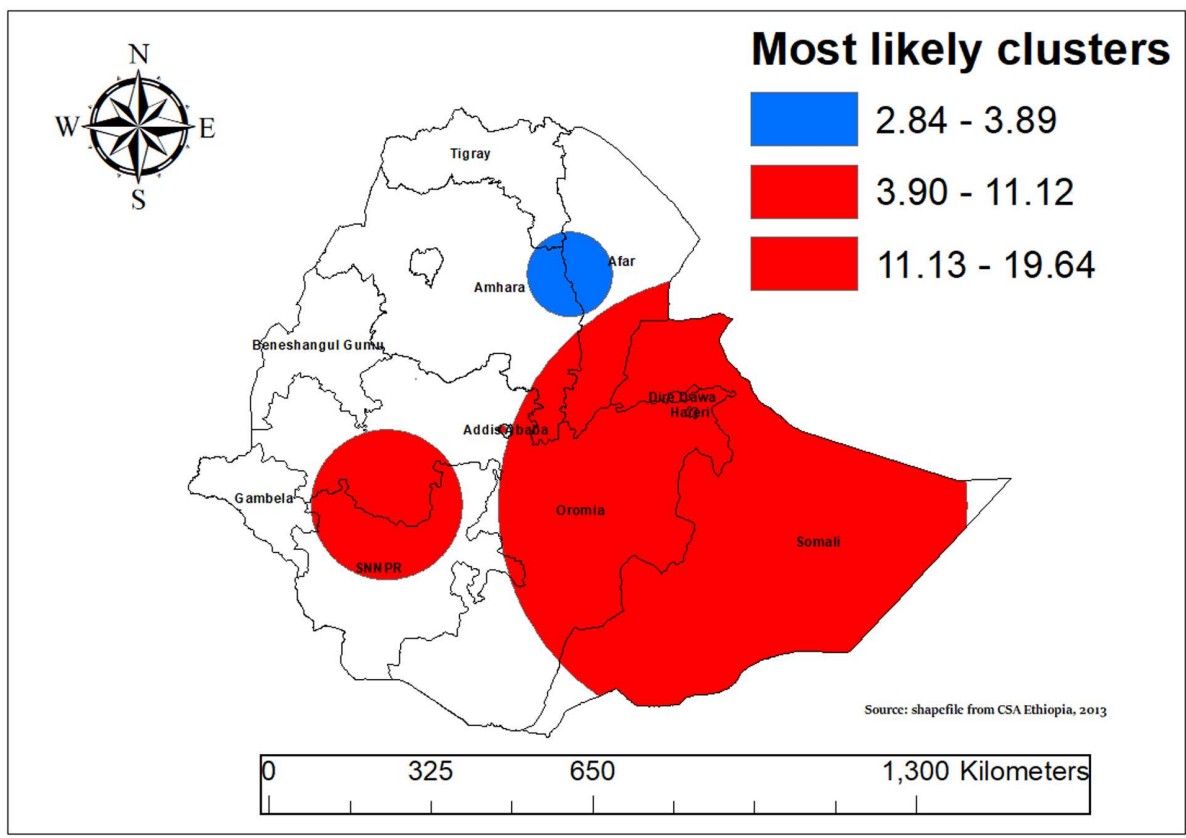

**Fig 7. SaTScan analysis of animal source food (ASF) consumption among children aged 6–23 months in Ethiopia, 2019.**

We examined the spatial patterns in the residuals to validate our GWR analysis results. The Moran's I statistic for the residuals from the GWR model analyzing ASF consumption among children aged 6–23 months yielded a Moran's I value of -0.075 with a p-value of 0.150. This indicates that the residuals are randomly distributed without significant spatial clustering or dispersion. These results suggest that the GWR model effectively captured the spatial patterns in ASF consumption and that the variables included in the model adequately explained the spatial dependencies observed in the residuals from the global OLS model (Fig 8).

The GWR analysis shows that ASF consumption among children aged 6–23 months in Ethiopia is notably influenced by several factors. Specifically, households with two or fewer children, those in the richest wealth index, and mothers with secondary or higher education levels are significantly associated with higher ASF consumption. Additionally, being of the Orthodox religion is also a significant factor. The impact of these factors varies across different regions of the country.

Mother's education (both secondary and higher) positively influenced children's ASF intake in our final GWR model, with coefficients ranging from 0.211 to 0.609. This indicates that a 1% increase in the percentage of mothers with higher education is associated with a 21.1% to 60.9% increase in ASF intake among children. The effect was significant throughout most of Ethiopia, with the strongest increases observed in Tigray, Amhara, Afar, and northern Somali (Fig 9).

For the Orthodox religion, the coefficient in our final GWR model ranges from -0.272 to 0.046. This generally means that being Orthodox tends to be linked with lower ASF intake among children, though in some areas, the effect can be slightly positive. As the proportion of Orthodox followers increased, ASF consumption decreased significantly in regions such as Afar, Amhara, some parts of Oromia, and Tigray (Fig 10).

**Table 3. Summary of OLS results ASF consumption among children aged 6-23 months in Ethiopia, 2019.**

| Variable | Coefficient | StdError | t-Statistic | Probability | Robust_SE | Robust_t | Robust_Pr | VIF |
|---|---|---|---|---|---|---|---|---|
| Intercept | 0.32 | 0.22 | 1.47 | 0.14 | 0.25 | 1.28 | 0.20 | -------- |
| Maternal age 25–34 | 0.13 | 0.07 | 1.76 | 0.08 | 0.08 | 1.54 | 0.12 | 1.19 |
| Urban residence | -0.02 | 0.07 | -0.34 | 0.73 | 0.06 | -0.39 | 0.70 | 3.45 |
| Mothers' secondary school | 0.21 | 0.11 | 1.99 | 0.04* | 0.10 | 2.06 | 0.04* | 1.28 |
| Mothers' higher education | 0.31 | 0.11 | 2.71 | <0.001* | 0.09 | 3.32 | <0.001* | 1.75 |
| Orthodox | -0.12 | 0.06 | -2.06 | 0.04* | 0.06 | -2.09 | 0.03* | 2.10 |
| Muslim | 0.07 | 0.05 | 1.32 | 0.19 | 0.06 | 1.29 | 0.20 | 2.11 |
| | | | | | | | | |
| Households with ≤2 Children | 0.16 | 0.08 | 2.03 | 0.04* | 0.08 | 1.98 | 0.04* | 1.24 |
| Richer wealth index | 0.05 | 0.08 | 0.64 | 0.52 | 0.09 | 0.58 | 0.56 | 1.48 |
| Richest wealth index | 0.17 | 0.09 | 1.92 | 0.06 | 0.08 | 2.09 | 0.03* | 4.79 |
| currently no breast feeding | 0.15 | 0.10 | 1.53 | 0.13 | 0.10 | 1.46 | 0.15 | 1.38 |
| Married | 0.02 | 0.13 | 0.12 | 0.90 | 0.15 | 0.10 | 0.92 | 1.12 |
| Birth order ≤ 3 | 0.07 | 0.08 | 0.88 | 0.38 | 0.08 | 0.88 | 0.38 | 1.63 |
| Sex of child(Female) | -0.04 | 0.07 | -0.62 | 0.54 | 0.07 | -0.58 | 0.57 | 1.08 |
| Child age 12–17 months | 0.00 | 0.08 | -0.04 | 0.97 | 0.09 | -0.04 | 0.97 | 1.46 |
| Child age 18–23 months | -0.03 | 0.09 | -0.34 | 0.73 | 0.09 | -0.34 | 0.74 | 1.60 |
| Had ANC follow-up | -0.14 | 0.10 | -1.48 | 0.14 | 0.12 | -1.22 | 0.22 | 2.62 |
| Health facility delivery | 0.11 | 0.09 | 1.28 | 0.20 | 0.09 | 1.25 | 0.21 | 3.71 |
| No Cesarean delivery | -0.09 | 0.12 | -0.74 | 0.46 | 0.12 | -0.79 | 0.43 | 1.48 |
| Getting counseling during pregnancy | -0.02 | 0.07 | -0.37 | 0.71 | 0.07 | -0.36 | 0.72 | 1.95 |

| OLS Diagnostics | | | | |
|---|---|---|---|---|
| Number of Observations: | 294 | AICc | | 140.89 |
| Multiple R-Squared | 26 | Adjusted R-Squared | | 21 |
| Joint F-Statistic | 5.07 | Prob(>F), (19,274) degrees of freedom | | <0.001* |
| Joint Wald Statistic | 176.52 | Prob(>chi-squared), (19) df | | <0.001* |
| Koenker (BP) Statistic | 45.43 | Prob(>chi-squared), (19) df | | <0.001* |
| Jarque-Bera Statistic | 4.66 | Prob(>chi-squared), (2) df | | 0.09 |

AICc, Akaike's information criterion;

OLS, Ordinary least squares; ANC, Antenatal care; VIF, Variance Inflation Factor

Households with two or fewer children tend to consume more ASF in many parts of Ethiopia. This is especially true in regions like Tigray, Afar, Amhara, eastern Oromia, and parts of Somali. In these areas, having fewer children in the household is linked to higher ASF consumption, indicating that smaller households are more likely to include these foods in their diet (Fig 11).

Households in the wealthiest category tend to consume more ASF across nearly all of Ethiopia. This is especially true in regions like SNNPR, Amhara, parts of Oromia, Addis Ababa, south western Afar and Benishangul-Gumuz. In these areas, having a higher wealth index often means greater access to and consumption of ASF. This pattern shows that wealthier households are more likely to include these foods in their diets, a trend that is seen throughout much of the country(Fig 12).

## Discussion

The aim of this study was to evaluate the spatial pattern and factors influencing the consumption of ASF among children aged 6–23 months old in Ethiopia. The prevalence of ASF consumption was 47.7% (95% CI: 45.1% - 50.2%),

**Table 4. GWR Model Results for Analyzing Spatial Patterns of Animal Source Food Consumption among children aged 6.-23 months in Ethiopia, 2019.**

| Explanatory variable | Households belonging to middle wealth index and baby postnatal check | |
|---|---|---|
| Residual Squares | 21.93 | |
| Effective Number | 23.57 | |
| Sigma | 0.28 | |
| AICc | 110.67 | |
| R2 | 32 | |
| R2Adjusted | 26 | |
| **Model comparison** | | |
| Parameters | OLS model | GWR model |
| AICc | 140.89 | 110.67 |
| Multiple R2 | 26 | 32 |
| Adjusted R2 | 21 | 26 |

AICc, Akaike's information criterion; EDHS, Ethiopian Demographic and Health Survey; GWR, geographically weighted regression; OLS, ordinary least squares

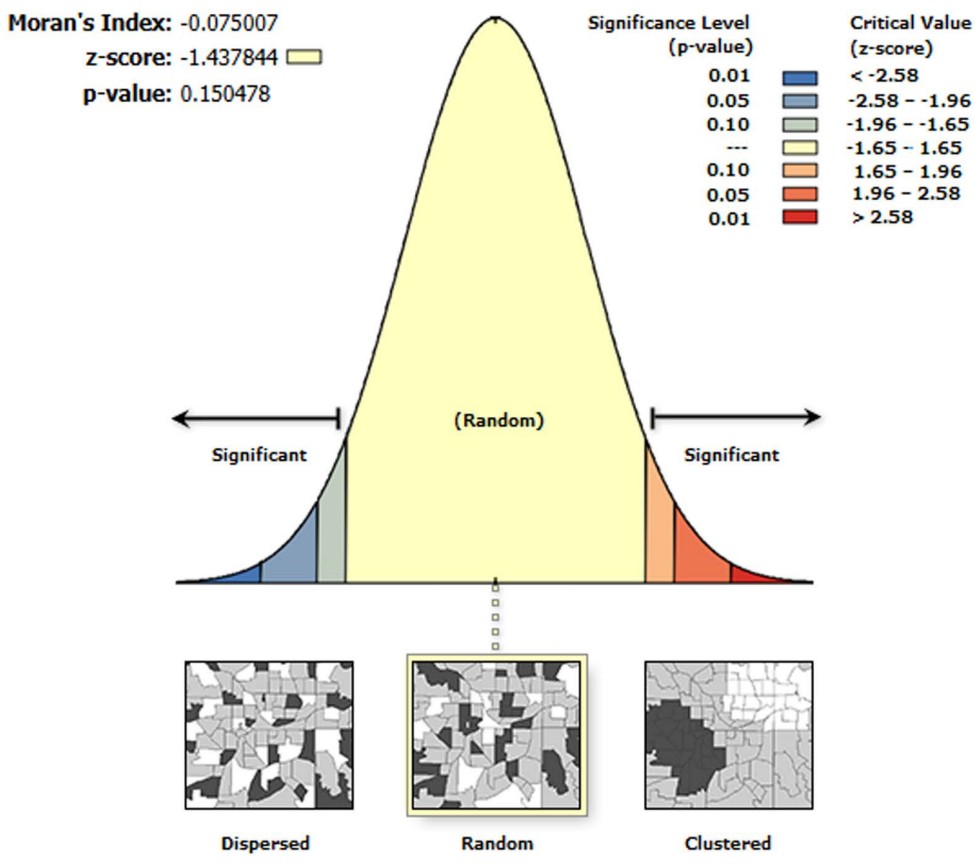

Given the z-score of -1.43784394234, the pattern does not appear to be significantly different than random.

**Fig 8. Moran's I statistic for the residuals of the geographically weighted regression (GWR) model on animal source food (ASF) consumption among children aged 6–23 months in Ethiopia, 2019.**

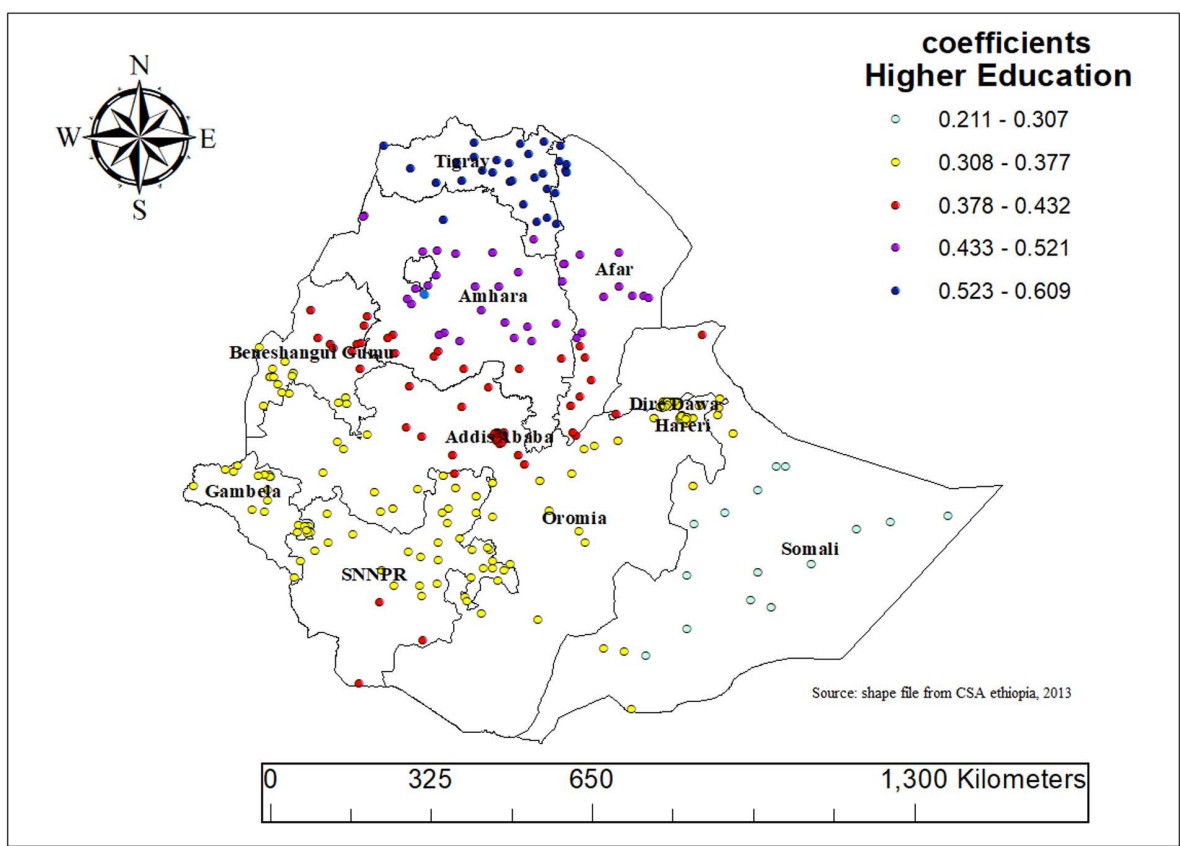

**Fig 9. Geographically weighted regression (GWR) coefficients showing spatial variation in the effect of maternal education on animal source food (ASF) consumption among children aged 6–23 months in Ethiopia, 2019.**

highlighting a consistent clustering of both high and low consumption areas among children aged 6–23 months in Ethiopia. Several important factors were found to be associated with increased consumption of ASF, such as mothers with higher levels of education, households with a smaller number of children, and families belonging to the highest wealth index. Conversely, a greater percentage of individuals following the Orthodox religion was correlated with reduced consumption of ASF.

The prevalence of ASF consumption (47.7%) among children aged 6–23 months in Ethiopia was consistent with the studies done in Ethiopia which showed that ASF prevalence was 48.3%, 46.5% respectively [27,28]. On the other hand the finding of this study was higher than the study conducted in sub-Saharan African countries which showed the prevalence of ASF consumption was 43.1% [29]. Several factors may contribute for this discrepancy, including differences in dietary practices, cultural preferences, socioeconomic status, and access to ASF across different regions. Moreover, local nutritional interventions and programs in Ethiopia may have contributed to the higher prevalence of ASF consumption observed in this study. Ethiopia has adopted the Food-Based Dietary Guidelines, the National Nutrition Program, the Food and Nutrition Policy, and the National Food and Nutrition Strategy as comprehensive measures to improve dietary practices, combat under-nutrition, and promote the consumption of animal source foods [30,31]. The prevalence of ASF consumption observed in this study is higher than previous DHS surveys in Ethiopia, which reported ASF consumption prevalence rates of 31.3% in 2005, 35.9% in 2011, and 41.5% in 2016 [32]. This increase might be due to improvements in nutritional programs and growing awareness about the importance of ASF in children's diets could play a role. Additionally, better socioeconomic

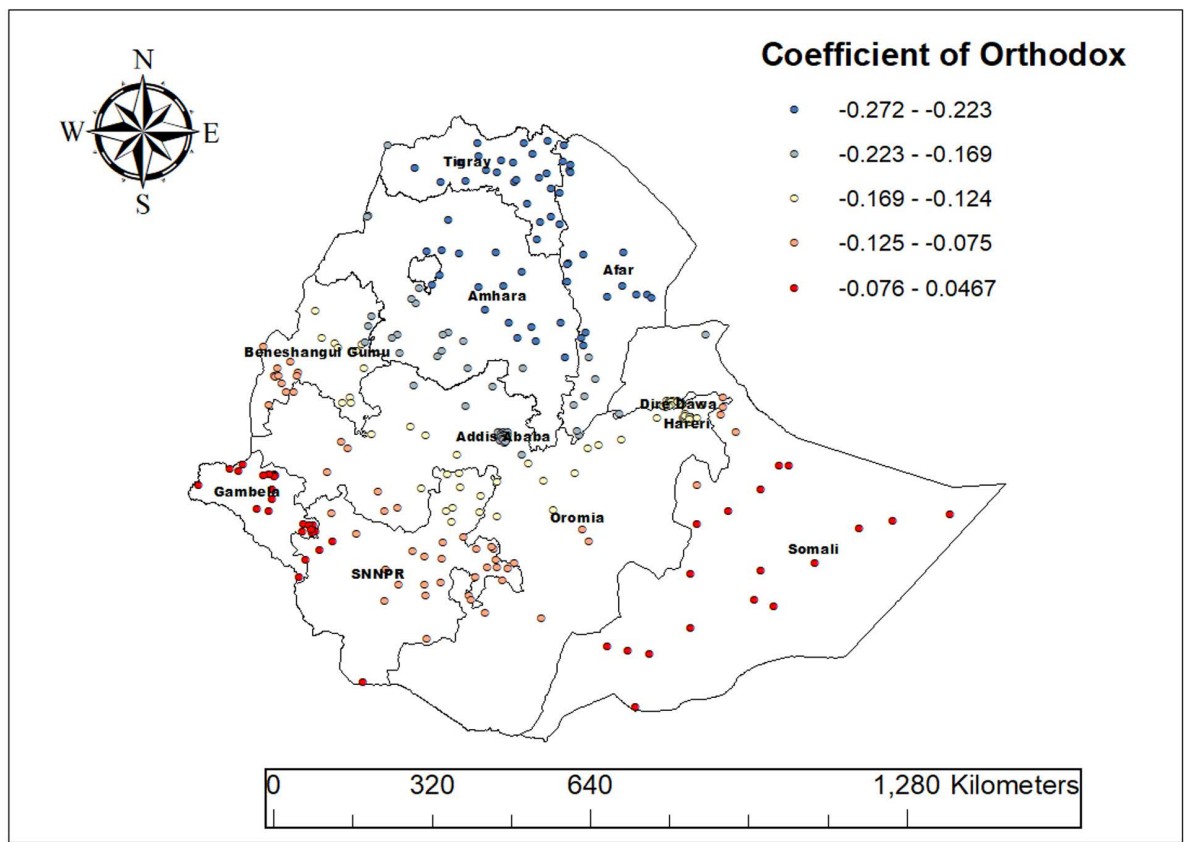

**Fig 10. Geographically weighted regression (GWR) coefficients showing spatial variation in the effect of Orthodox religion on animal source food (ASF) consumption among children aged 6–23 months in Ethiopia, 2019.**

conditions and more accessible ASF might also be contributing to this trend. Moreover, targeted interventions and government policies focused on boosting child nutrition may have further supported this rise in ASF consumption.

The study found that children from orthodox Christian follower had decrease the consumption of ASF. This finding is consistent with the previous studies [28,33]. The doctrine of the Ethiopian Orthodox Church includes numerous fasting days throughout the year, such as most Wednesdays and Fridays. Additionally, there are extended fasting periods like the Great Lent Fast, which lasts 55 days, and the Fast of the Prophets, which lasts 43 days [33,34]. During these times, adults abstain from ASF, including eggs and dairy products. Even though fasting practices in the Ethiopian Orthodox Church primarily target adults, children's reduced consumption of ASF during these periods can be attributed to family dietary adjustments, parental influence, resource reallocation, and cultural norms extending to children's diets.

The current study found that higher levels of maternal education were significantly associated with increased ASF consumption among children. This finding aligns with several studies conducted in Ethiopia. For instance, a study in the Ethiopia found that children of mothers with secondary or higher education were more likely to consume ASF compared to those whose mothers had no formal education [32,35]. Another study in sub-Saharan country reported similar results, indicating that maternal education positively influences child nutrition by increasing knowledge about healthy feeding practices and access to diverse food options [29]. A study in China also supports this association, finding that maternal education was a strong predictor of better dietary diversity, including higher ASF intake among young children [36]. Specifically,

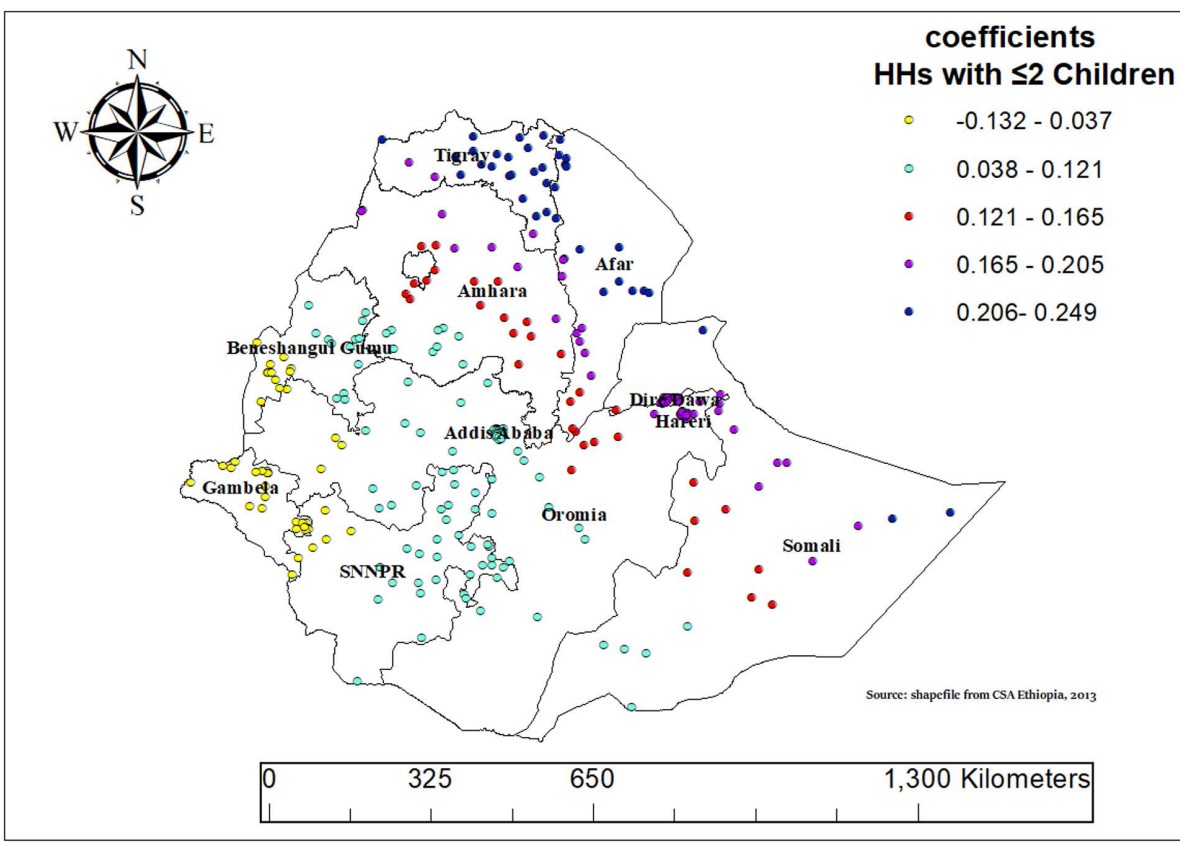

**Fig 11. Geographically weighted regression (GWR) coefficients showing spatial variation in the effect of households with ≤2 children on animal source food (ASF) consumption among children aged 6–23 months in Ethiopia, Mini-EDHS, 2019.**

educated mothers are more likely to understand the nutritional benefits of ASF and have better access to resources that facilitate the provision of a diverse diet for their children [37].

This study revealed that households with a smaller number of children (two or fewer) were more likely to have higher ASF consumption among children. This observation aligns with findings from various studies that indicate smaller household size often correlates with better child nutrition [38,39]. Household size, as a significant socioeconomic determinant, has been shown to affect food security with larger households often facing reduced food acquisition and lower per capita consumption of animal source foods (ASF) due to greater resource constraints [40,41]. Animal source foods are expensive making it challenging for larger families to afford enough for everyone. In larger households, the financial strain often means that ASF is sold to cover other essential family needs, which reduces the amount available for children to consume.. Therefore, the cost of ASF and its impact on family finances play a crucial role in the differences in ASF consumption between small and large households.

This study showed that families belonging to the highest wealth index were significantly more likely to have higher ASF consumption among children. This association has been documented in multiple studies. In a study from Ethiopia, higher household wealth was associated with increased consumption of meat, eggs, and dairy products among children [22]. Another study from sub-Saharan Africa also found that children from wealthier families had a significantly higher likelihood of consuming ASF [29]. Research from Nepal corroborates these findings, indicating that higher household wealth is associated with increased consumption of flesh foods and dairy products among children [42]. Wealthier households can afford

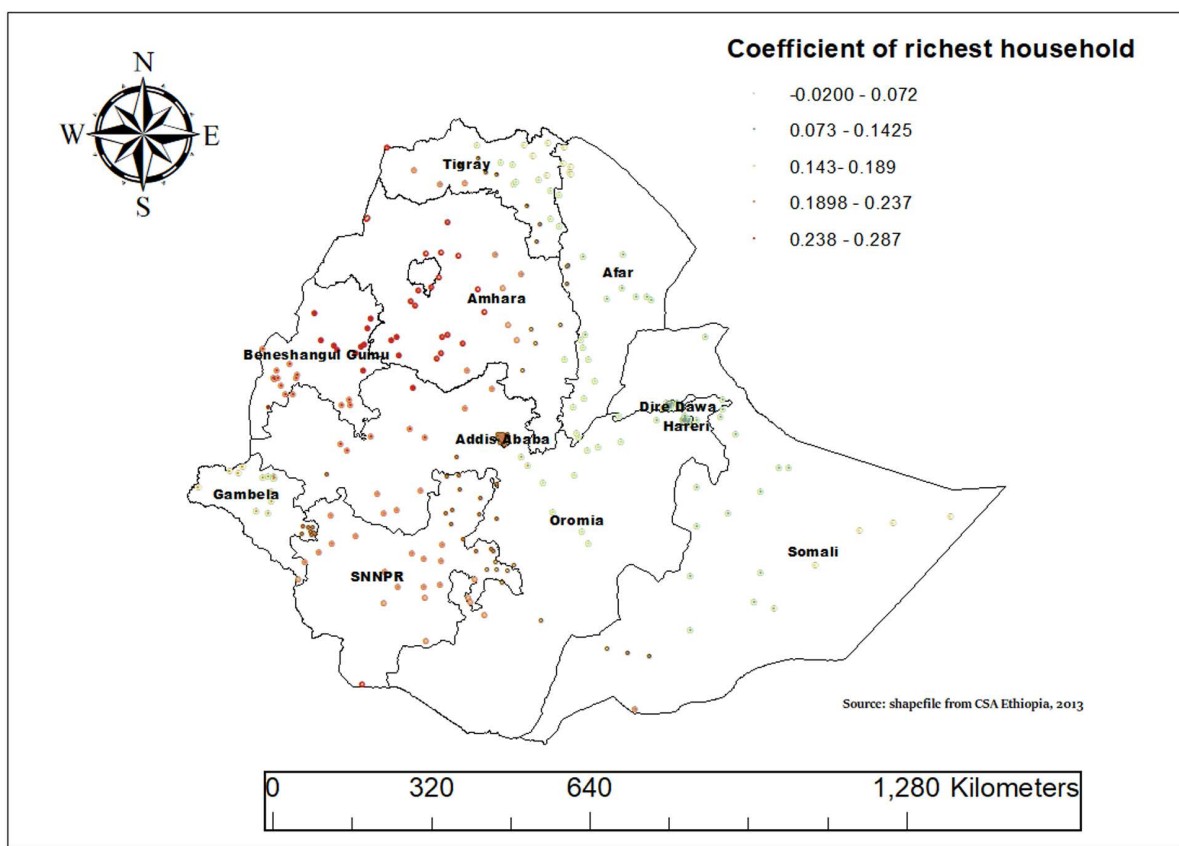

**Fig 12. Geographically weighted regression (GWR) coefficients showing spatial variation in the effect of the richest wealth index on animal source food (ASF) consumption among children aged 6–23 months in Ethiopia, Mini-EDHS, 2019.**

a more diverse diet, including more expensive items like ASF. Their wealth allows them access to a variety of foods and reduces the risk of food insecurity, leading to better nutritional outcomes. Additionally, these households often have better access to markets, making it easier for them to purchase a range of foods, including ASF [43].

### Strengths and limitations of this study

This study has several strengths. Firstly, it uses the 2019 EDHS dataset, which is nationally representative and enhances the generalizability of the findings. Secondly, the study includes spatial analysis, which helps account for regional variations in ASF consumption. However, this study has a few limitations. The data from the Mini-EDHS 2019 are cross-sectional, so causality cannot be established. Additionally, self-reported dietary data may introduce recall bias. Some relevant factors, like household food access, may not be fully captured in the dataset. Lastly, while the GWR model highlights regional variations, unmeasured confounders could still influence the findings.

### Conclusion and recommendation

The study reveals that 47.7% of Ethiopian children aged 6–23 months consume ASF which is relatively low and with notable clustering of low consumption areas in regions such as Amhara, Tigray, Benishangul-Gumuz, western SNNPR, and Gambela region. This clustering indicates significant regional variations in ASF consumption among this demographic.

Key factors associated with higher ASF consumption include maternal education, smaller household size, a higher proportion of Orthodox religion followers, and greater household wealth. These findings underscore the influence of socioeconomic and cultural factors on ASF consumption patterns.

To improve ASF consumption, we should focus on educating mothers about the benefits of ASFs, especially in areas where intake is low. Reducing family sizes through better access to family planning can help families afford more nutritious foods. Supporting economic growth will also improve access to ASFs. Programs need to be tailored to local needs, with subsidies making these foods more affordable, and involving communities in designing solutions. Working with private and public sectors to strengthen food supply chains and integrating ASF promotion into healthcare will help ensure lasting improvements in nutrition. Policymakers and public health practitioners should prioritize maternal education, family planning, and poverty alleviation strategies, especially in identified low-consumption areas, to improve ASF intake and child nutrition outcomes nationally.

## Acknowledgments

We acknowledge The DHS Program for granting us access to utilize the Ethiopia Demographic and Health Survey (EDHS) data for our analysis.

## Author contributions

**Conceptualization:** Mekuriaw Nibret Aweke.

**Data curation:** Mekuriaw Nibret Aweke, Gebrie Getu Alemu, Berihanu Mengistu.

**Formal analysis:** Mekuriaw Nibret Aweke, Amare Mesfin, Gebrie Getu Alemu, Berihanu Mengistu, Tewodros Getaneh Alemu, Habtamu Wagnew Abuhay.

**Funding acquisition:** Amare Mesfin, Habtamu Wagnew Abuhay.

**Investigation:** Mekuriaw Nibret Aweke, Amare Mesfin, Berihanu Mengistu, Tewodros Getaneh Alemu, Habtamu Wagnew Abuhay.

**Methodology:** Mekuriaw Nibret Aweke, Amare Mesfin, Tewodros Getaneh Alemu, Habtamu Wagnew Abuhay.

**Project administration:** Mekuriaw Nibret Aweke.

**Resources:** Gebrie Getu Alemu, Tewodros Getaneh Alemu, Habtamu Wagnew Abuhay.

**Software:** Mekuriaw Nibret Aweke, Amare Mesfin, Berihanu Mengistu.

**Supervision:** Mekuriaw Nibret Aweke, Amare Mesfin.

**Validation:** Mekuriaw Nibret Aweke, Amare Mesfin.

**Visualization:** Mekuriaw Nibret Aweke, Gebrie Getu Alemu, Tewodros Getaneh Alemu.

**Writing – original draft:** Mekuriaw Nibret Aweke.

**Writing – review & editing:** Mekuriaw Nibret Aweke.

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
