## [Decision Letter · Decision Letter 0]

17 Apr 2025

PONE-D-24-34120Spatial Distribution of Animal Source Food Consumption and Associated Factors Among Children Aged 6–23 Months in Ethiopia: A Geographically Weighted Regression AnalysisPLOS ONE

Dear Dr. Aweke,

Thank you for submitting your manuscript to PLOS ONE. After careful consideration, we feel that it has merit but does not fully meet PLOS ONE’s publication criteria as it currently stands. Therefore, we invite you to submit a revised version of the manuscript that addresses the points raised during the review process.

Based on the analysis of the three reviewers' reports, below is an editorial synthesis indicating **required changes for acceptance** and **recommended changes** : **Required Changes for Acceptance:** 

**Abstract and Conclusion** : Clarify the main contributions and practical implications of the study.**Introduction** : Improve the logical flow of paragraphs and refine redundant statements.**Justification and Methods** : Strengthen the rationale for the study focus and analytical approach; clarify data preprocessing and variable treatment; ensure consistent notation and define all abbreviations at first mention.**Results and Figures** : Correct table percentages, improve figure references and readability, and standardize formatting and terminology.**Discussion** : Address study limitations (e.g., data source), expand contextual discussion (e.g., existing programs and socioeconomic aspects), and streamline content by removing redundant sections.**Recommendations** : Provide more specific, actionable recommendations based on the findings.

<h3 data-end="1034" data-start="1006">**Recommended Changes:** </h3>

Consider refining the writing style in the discussion to improve fluency.Explore potential correlations among independent variables and comment on their implications.If feasible, consider performing geospatial analysis at a more granular level for greater policy relevance.

These revisions are necessary to meet PLOS ONE’s criteria for methodological rigor, clarity, transparency, and contextual relevance.

We look forward to receiving your revised manuscript.

Kind regards,

Elma Izze Da Silva Magalhães

Academic Editor

PLOS ONE

2. We note that your Data Availability Statement is currently as follows: “All relevant data are within the manuscript and its Supporting Information files”

Please confirm at this time whether or not your submission contains all raw data required to replicate the results of your study. Authors must share the “minimal data set” for their submission. PLOS defines the minimal data set to consist of the data required to replicate all study findings reported in the article, as well as related metadata and methods (https://journals.plos.org/plosone/s/data-availability#loc-minimal-data-set-definition ).

If your submission does not contain these data, please either upload them as Supporting Information files or deposit them to a stable, public repository and provide us with the relevant URLs, DOIs, or accession numbers. For a list of recommended repositories, please see https://journals.plos.org/plosone/s/recommended-repositories .

3. We note that Figures 4, 5, 6, 7, 9, 10, 11, 12 and 13 in your submission contain [map/satellite] images which may be copyrighted. All PLOS content is published under the Creative Commons Attribution License (CC BY 4.0), which means that the manuscript, images, and Supporting Information files will be freely available online, and any third party is permitted to access, download, copy, distribute, and use these materials in any way, even commercially, with proper attribution. For these reasons, we cannot publish previously copyrighted maps or satellite images created using proprietary data, such as Google software (Google Maps, Street View, and Earth). For more information, see our copyright guidelines: http://journals.plos.org/plosone/s/licenses-and-copyright .

   a. You may seek permission from the original copyright holder of Figures 4, 5, 6, 7, 9, 10, 11, 12 and 13 to publish the content specifically under the CC BY 4.0 license.   

Reviewers' comments:

Reviewer's Responses to Questions

**Comments to the Author**

1. Is the manuscript technically sound, and do the data support the conclusions?

Reviewer #1: Yes

Reviewer #2: Yes

Reviewer #3: Yes

2. Has the statistical analysis been performed appropriately and rigorously? 

Reviewer #1: Yes

Reviewer #2: Yes

Reviewer #3: Yes

3. Have the authors made all data underlying the findings in their manuscript fully available?

Reviewer #1: Yes

Reviewer #2: Yes

Reviewer #3: Yes

4. Is the manuscript presented in an intelligible fashion and written in standard English?

Reviewer #1: Yes

Reviewer #2: Yes

Reviewer #3: Yes

5. Review Comments to the Author

Reviewer #1: The paper addresses the topic of child nutrition and utilizes a geospatial technique that is increasingly used in public health. It shows originality and makes a significant contribution to the topic. The empirical strategies are well explained. However, there are a few areas that could be revised to enhance the content of the manuscript.

1. Conclusion in the abstract should be redrafted to highlight the contribution of the article and/or specific implication for practice or methodology.

2. A strong rationale for focusing on animal food source consumption of children aged 6-23 months is missing. Additionally, consider adding some strong justification for why the authors employed a geographically weighted regression technique to identify associated factors.

3. Please write acronyms/abbreviations in the long term when used for the first time. There are places where the abbreviation is used before it is introduced.

4. A clear explanation of the data preprocessing is required. E.g. how do you handle missing values in the dependent variable and independent variables?

5. Percentages in Table One should be revised and need to be calculated properly.

6. The spatial analysis was done at the regional level. I think it would be better to do it at the sub-national level (preferably at the Zonal or woreda level).

7. More appropriate recommendations and precise comparisons of determinants in this study relative to existing literature, e.g., (such as religion were found to be significant in this study), are expected to aid in the understanding and interest of readers on this important topic.

8. It can be noted that the dependent variable of ASF came from self-report and this would undermine validity and reliability. I would like to ask the authors to address this issue in the section of the Discussion.

Reviewer #2: The article is interesting and well structured. The rationale is justified, and the methodology is solid and adequate. Some results, such as the Mini-EDHS diagram (Figure 1), the Moran’s Index graphs (Figures 2 and 8) and the tables, are very well presented and self-explanatory. However, it requires a few minor adjustments.

Abstract: Put the acronym ASF adjacent to "Animal source foods" in "Animal source foods provide essential nutrients like high-quality protein, iron, zinc, calcium, and vitamin B12, which are vital for the physical and cognitive development of young children.", please. It will make the abstract easier to read.

Methodology:

- Geographically weighted regression analysis: Check the writing of the variables, there are differences between the equation and the text. For example, the equation presents two variables as "Bk (ui, vi)" and "Ei", while they are written in the text as "βk (ui, vi)" and "εii".

- Independent Variables: Have you tested a possible correlation between the sociodemographic characteristics and the socioeconomic factors considered (wealth, education and children per household)? For example, families with higher education may tend to be richer and to have less children. If they are highly correlated, presenting an analysis of each variable separately might be redundant.

Results:

1. In the "Spatial autocorrelation by distance" graph (Figure 3), please put the distance in kilometers instead of meters. The same applies to the related paragraph. It is simpler for the reader to understand 300 km than 300,000 m.

2. In the maps (Figures 4 to 7 and 9 to 13), some areas’ names are difficult to read. Maybe try to adjust the points’ or polygons’ (generated by the analysis, not the base polygon with the areas) opacity to make it more transparent.

3. In Figure 4, change the color of the "non significant" results to make it different from the "Hot Spots". In Figure 5, for instance, this issue is nonexistent.

4. In Figure 7, change the legend title from "Most likly clusters" to "Most likely clusters".

5. In Figure 11 ("figure 11.tif"), review the color palette because the first two categories ("-0.272 to -0.223" and "-0.223 to -0.169") have similar colors.

6. Please check the reference to the figures from Figure 9 to 13 in the text. The "Figure 9" in the text appears to be referring to Figure 10 ("Figure he 10.tif"), while “Figure 10” is referring to Figure 11 ("figure 11.tif"), etc. And I did not understand to which part of the text the "figure 9 sec.tif" file is related.

Reviewer #3: General Considerations

The article is well-written and presents a strong line of reasoning based on the literature. The methodology is well-structured and ensures replicability, which enhances the study’s reliability. The selected analyses are appropriate for the study’s objective and demonstrate methodological rigor in data handling and interpretation. The comments and suggestions below aim to improve the manuscript and enhance the quality of the work, with a view to publication in the journal.

Specific Comments:

Title:

Appropriate and effectively conveys the study’s purpose

Abstract:

Well-written and cohesive, clearly presenting the research problem. The recommendations at the end could be refined for greater clarity and impact.

The introduction presents and connects relevant facts related to the theme, justifying the need for the study. However, reorganizing the paragraphs could improve the flow and clarity of the argument. For instance, the paragraph discussing Ethiopia's malnutrition panorama would be more logically placed immediately after the global and continental data. The paragraph on the consumption of animal-source foods, currently positioned between them, disrupts the continuity of the discussion.

A more coherent sequence might be:

The importance of nutrition in the first two years of life (paragraph 1);

The issue of malnutrition at the global level and in Sub-Saharan Africa (paragraph 2);

The malnutrition situation in Ethiopia, supported by research data (paragraph 4);

Dietary consumption patterns in Ethiopia, directly linked to the previous paragraph (paragraph 5).

Additionally, revising the final sentence of paragraph 3 would help avoid repetition. The closing paragraph effectively summarizes the introduction, providing a clear and consistent justification for the study.

Methods:

The methods section is well-structured and provides a detailed description of the strategies employed, especially regarding statistical procedures. The techniques used are appropriately justified and align well with the study’s objectives, demonstrating methodological rigor. The clear presentation of methods enhances both comprehension and replicability, adding transparency and robustness to the research.

Results:

The results are well-organized and clearly presented, effectively supporting the subsequent discussion. The information is structured to facilitate understanding and logical progression. However, a careful review of the figure numbering is recommended to ensure alignment with the order in which they are referenced in the text. In the reviewed file, some figures appeared out of sequence, which could lead to confusion for the reader. Additionally, improving the visual quality of the figures would enhance the graphical representation of the data.

Discussion:

The discussion thoroughly addresses all findings, contextualizing them within existing literature, which reinforces theoretical grounding and consistency. However, the repetitive structure of the paragraphs—each following a similar format—makes the reading somewhat monotonous. A rewrite is suggested to improve fluency and dynamism, creating a more engaging and cohesive reading experience.

Furthermore, the authors briefly mention the existence of nutritional programs in Ethiopia but do not provide details or references. Expanding on this aspect would significantly enrich the discussion and better contextualize the findings. Similarly, it would be valuable to elaborate on socioeconomic factors, particularly household size and its potential influence on food acquisition. Incorporating national surveys or studies that collect this type of data could offer relevant insights into food insecurity and dietary intake in the studied context.

Finally, removing the last paragraph of the discussion is recommended, as it reiterates points already covered earlier in the section and in the introduction. Its omission would contribute to a more concise and focused discussion. The section on limitations and strengths is excellent, reinforcing the methodological rigor applied throughout the manuscript.

Conclusion:

The conclusion effectively addresses the study’s objective, summarizing key findings clearly. However, the recommendations based on the results could be made more explicit. What existing policies or programs aimed at improving child nutrition in Ethiopia could be refined in light of the findings? Additionally, it would be beneficial to discuss the essential elements a new public policy should incorporate to promote increased consumption of animal-source foods among children, considering relevant social, economic, and cultural challenges.

6. PLOS authors have the option to publish the peer review history of their article (what does this mean? ). If published, this will include your full peer review and any attached files.

**Do you want your identity to be public for this peer review?** For information about this choice, including consent withdrawal, please see our Privacy Policy .

Reviewer #1: No

Reviewer #2: No

Reviewer #3: No

---

## [Author Response · Author response to Decision Letter 1]

28 Apr 2025

we uploaded the response to reviewer in word document

---

## [Editor Report · Decision Letter 1]

2 May 2025

Spatial Distribution of Animal Source Food Consumption and Associated Factors Among Children Aged 6–23 Months in Ethiopia: A Geographically Weighted Regression Analysis

PONE-D-24-34120R1

Dear Dr. Aweke,

We’re pleased to inform you that your manuscript has been judged scientifically suitable for publication and will be formally accepted for publication once it meets all outstanding technical requirements.

Kind regards,

Elma Izze Da Silva Magalhães

Academic Editor

PLOS ONE

---

## [Editor Report · Acceptance letter]

PONE-D-24-34120R1

PLOS ONE

Dear Dr. Aweke,

I'm pleased to inform you that your manuscript has been deemed suitable for publication in PLOS ONE. Congratulations! Your manuscript is now being handed over to our production team.

Kind regards,

on behalf of

Dr. Elma Izze Da Silva Magalhães

Academic Editor

PLOS ONE